# Comparing the Outcomes of Ceftaroline plus Vancomycin or Daptomycin Combination Therapy versus Vancomycin or Daptomycin Monotherapy in Adults with Methicillin-Resistant *Staphylococcus aureus* Bacteremia—A Meta-Analysis

**DOI:** 10.3390/antibiotics11081104

**Published:** 2022-08-15

**Authors:** Chienhsiu Huang, Ihung Chen, Lichen Lin

**Affiliations:** 1Department of Internal Medicine, Dalin Tzu Chi Hospital, Buddhist Tzu Chi Medical Foundation, Chiayi 62247, Taiwan; 2Department of Nursing, Dalin Tzu Chi Hospital, Buddhist Tzu Chi Medical Foundation, Chiayi 62247, Taiwan

**Keywords:** vancomycin, daptomycin, ceftaroline, *Methicillin-resistant Staphylococcus aureus*, bacteremia

## Abstract

Introduction: Combination therapy with daptomycin plus ceftaroline to treat methicillin-resistant *Staphylococcus aureus* bacteremia has been reported to reduce methicillin-resistant *Staphylococcus aureus* bacteremia-related mortality. The purpose of the current meta-analysis was to compare the clinical outcome of methicillin-resistant *Staphylococcus aureus* bacteremia in patients treated with daptomycin or vancomycin plus ceftaroline combination therapy versus daptomycin or vancomycin monotherapy. Methods: Studies were included if they directly compared the efficacy of daptomycin or vancomycin plus ceftaroline combination therapy with that of daptomycin or vancomycin monotherapy in the treatment of methicillin-resistant *Staphylococcus aureus* bacteremia in adult patients. Results: One randomized controlled trial and five retrospective studies were included in the meta-analysis. The combination therapy group had an in-hospital mortality, duration of bacteremia, and adverse event rate similar to those patients who had monotherapy. There was less bacteremia recurrence in the combination group. Initial combination therapy with ceftaroline for the treatment of methicillin-resistant *Staphylococcus aureus* bacteremia showed a trend of reducing the risk of in-hospital mortality in the current meta-analysis. Conclusions: Randomized controlled trials are needed to further study the role of initial combination therapy with daptomycin or vancomycin plus ceftaroline in the treatment of methicillin-resistant *Staphylococcus aureus* bacteremia.

## 1. Introduction

Methicillin-resistant *Staphylococcus aureus* (MRSA) bacteremia is a serious infectious disease associated with a high risk of mortality [1,2,3]. The primary parenteral therapy for MRSA infection is vancomycin. Treatment with vancomycin will clinically or microbiologically fail in invasive MRSA infections [4,5,6,7,8]. Daptomycin is an alternative first-line option that is reserved for MRSA bacteremia that has relapsed or persisted despite vancomycin treatment. However, the failure of daptomycin in the treatment of MRSA bacteremia is common, and nonsusceptibility has emerged [9,10,11]. The clinical practice guidelines by the infectious diseases society of America for the treatment of MRSA infections in adults and children recommends the following management strategies for persistent MRSA bacteremia and vancomycin treatment failure in adult patients: 1. If the isolate is susceptible, high-dose daptomycin in combination with another agent (such as a beta-lactam antibiotic) should be considered (B-III strength of recommendation). 2. If a reduced susceptibility to vancomycin and daptomycin are present, options may include quinupristindalfopristin, trimethoprim-sulfamethoxazole, linezolid, or telavancin (C-III strength of recommendation) [12]. Ceftaroline is a fifth-generation cephalosporin that is active against gram-positive pathogens such as *Staphylococcus aureus* (SA) and *Streptococcus pneumoniae* as well as their resistant strains (e.g., MRSA, vancomycin-resistant SA, and multidrug-resistant *Streptococcus pneumoniae*) [13]. The in vitro activity of ceftaroline is also active against common gram-negative pathogens such as *Escherichia coli*, *Klebsiella pneumoniae*, and *Haemophilus influenzae*; it is not active against extended spectrum beta-lactamase-producing gram-negative organisms [14]. Ceftaroline is the only commercially available beta-lactam with bactericidal activity against MRSA; it also increases the bactericidal effect of daptomycin by enhancing daptomycin binding to bacterial cell membranes [15,16,17]. Ceftaroline is currently approved in the United States for bacterial pneumonia and skin/soft tissue infections [18,19]. Observational studies have shown that ceftaroline may be effective in the treatment of MRSA bacteremia. However, studies comparing ceftaroline to standard-of-care therapy for MRSA bacteremia are limited, and ceftaroline is not United States Food and Drug Administration-approved for this indication. Thus, combination therapy with ceftaroline plus daptomycin is an option because both agents have an individual efficacy against MRSA [20,21,22]. In the study by Zasowski et al., ceftaroline was noninferior to daptomycin in the treatment of MRSA bacteremia [23]. Studies have shown that combination therapy, such as the combination of daptomycin and ceftaroline, can reduce the duration of MRSA bacteremia and mortality due to MRSA bacteremia [24,25,26,27,28]. Therefore, we performed a comprehensive and updated meta-analysis of the clinical outcomes associated with daptomycin or vancomycin plus ceftaroline combination therapy in MRSA bacteremia patients. The purpose of the current meta-analysis was to compare the clinical outcomes of the in-hospital mortality, recurrence of MRSA bacteremia, duration of bacteremia, and adverse events between patients treated with daptomycin or vancomycin plus ceftaroline combination therapy and daptomycin or vancomycin monotherapy.

## 2. Results

### 2.1. Characteristics of the Included Trials

The details of the study selection process are shown in Figure 1. The numbers of studies from the initial search results from PubMed, Web of Science, and the Cochrane Library were 110, 114, and 27, respectively. There were 105 duplicate articles. A total of 146 irrelevant studies were identified by reading the title and/or abstract. After excluding the duplicates and irrelevant studies, 23 potentially relevant articles remained. After a full-text article review, 17 articles were excluded because they lacked results comparing the outcomes of ceftaroline plus vancomycin or daptomycin combination therapy versus vancomycin or daptomycin monotherapy in adults with MRSA bacteremia. Finally, six studies were included in the meta-analysis [29,30,31,32,33,34]. The main characteristics of the six included studies are shown in Table 1. One was a randomized controlled trial (RCT), and five were retrospective observational studies. The infection sites were all multiple primary infection sites. All of the studies had a high risk of bias.

### 2.2. Efficacy and Safety Outcomes

There were 200 patients in the monotherapy group and 147 patients in the combination therapy group. Five studies involving 299 patients (177 receiving monotherapy, 122 receiving ceftaroline combination therapy) reported bacteremia recurrence. There was a statistically significant difference in the bacteremia recurrence between the patients treated with monotherapy and those treated with ceftaroline combination therapy (OR = 2.95, 95% CI= 1.22–7.15, *p* = 0.02, I^2^ = 6%) (Figure 2). All five studies reported that the combination therapy group had a lower bacteremia recurrence than the monotherapy group. Six studies involving 347 patients (200 receiving monotherapy, 147 receiving ceftaroline combination therapy) reported in-hospital mortality. There was no statistically significant difference in the in-hospital mortality between the patients treated with monotherapy and those treated with ceftaroline combination therapy (OR = 1.24, 95% CI= 0.66–2.33, *p* = 0.50, I^2^ = 53%) (Figure 3). Three studies favored monotherapy, and two studies favored combination therapy. Merrisette et al. found no deaths in either group [32]. We analyzed the two studies in which the combination therapy group had a better in-hospital mortality than the monotherapy group [30,31]. Geriak et al. showed that combination therapy had mortality benefits as an initial therapy within 72 h of the index culture in the treatment of MRSA bacteremia [30]. McCreary et al. found a lower all-cause mortality at day 30 in the combination therapy group than in those who received standard-of-care monotherapy, and there was a lower mortality in the patients who received daptomycin plus ceftaroline within 72 h of the index culture [31]. Four studies involving 159 patients (82 receiving monotherapy, 77 receiving ceftaroline combination therapy) reported adverse events. There was no statistically significant difference in the adverse events between the patients treated with monotherapy and those treated with ceftaroline combination therapy (OR= 0.59, 95% CI= 0.27–1.27, *p* = 0.18, I^2^ = 16%) (Figure 4). There were two cases of eosinophilic pneumonitis and three cases of elevated creatine phosphokinase in the combination therapy group. There was no case of eosinophilic pneumonitis but two cases of elevated creatine phosphokinase in the monotherapy group. Regarding the duration of bacteremia, four studies reported no statistically significant difference in the duration of bacteremia between the patients treated with monotherapy and those treated with ceftaroline combination therapy [29,30,32,33]. In the study of McCreary et al., the mean duration of bacteremia was 4.8 days in the standard-of-care group and 9.3 days in the daptomycin plus ceftaroline group (*p* < 0.001); following the switch to daptomycin plus ceftaroline, the mean duration of continued bacteremia was 3.3 days for the combination therapy. The longer duration of bacteremia was related to daptomycin plus ceftaroline as a salvage therapy, and there were 6.0 mean bacteremia days before switching to combination therapy [31]. A study by Johnson et al. showed that the mean duration of bacteremia was 5.0 days in the standard-of-care group and 9.0 days in the daptomycin plus ceftaroline group (*p* = 0.01). The total duration of bacteremia was significantly longer in the daptomycin plus ceftaroline group because the patients failed first-line therapy (after a median of six days) before daptomycin plus ceftaroline combination therapy. The longer duration of bacteremia was related to daptomycin plus ceftaroline as a salvage therapy [34]. We found that daptomycin plus ceftaroline as a salvage therapy was associated with a longer duration of bacteremia in both studies.

## 3. Methods

### 3.1. Data Search Strategy

The literature search was performed using the PubMed, Web of Science, and Cochrane Library databases in order to identify all included clinical studies and meta-analyses or systematic reviews on the topic from 1 January 2009. In the databases, we used the following search string: (ceftaroline OR vancomycin OR daptomycin) (bacteremia OR MRSA bacteremia OR methicillin-resistant *Staphylococcus aureus* bacteremia). We examined treatment studies that directly compared the outcomes of ceftaroline plus vancomycin or daptomycin combination therapy versus vancomycin or daptomycin monotherapy in adults with MRSA bacteremia and searched the relevant articles published from inception to 30 May 2022. Previously published systematic reviews and meta-analyses were reviewed to identify any additional studies that may have been missed in the primary literature search. Articles published in all languages were included.

### 3.2. Study Selection and Data Extraction

To determine the eligibility of the identified trial reports, each study was independently screened and reviewed for eligibility by two authors. After excluding duplicates, two investigators screened the titles and abstracts of all the studies retrieved to identify eligible records. After excluding irrelevant studies, all of the relevant articles were reviewed by reading the full texts to determine eligibility. Data regarding author, year of publication, country, study design, primary infection sites, total number of patients receiving monotherapy, total number of patients receiving combination therapy, antibiotic dosage, initial therapy or salvage therapy, in-hospital mortality, bacteremia recurrence, duration of bacteremia, and adverse events were extracted from the eligible full text articles. When disagreement occurred, a third author resolved the issue.

### 3.3. Inclusion and Exclusion Criteria

Due to inadequate levels of evidence, observational studies are not as meaningful as RCTs. There was a very small number of RCTs available. We included retrospective observational studies, prospective observational studies, and RCTs in the current meta-analysis. The studies were considered eligible for inclusion only if they directly compared the outcomes of ceftaroline plus vancomycin or daptomycin combination therapy versus vancomycin or daptomycin monotherapy in adults with MRSA bacteremia. Ceftaroline was administered at dosages ranging from 600 mg every 12 h to 600 mg every 8 h. Daptomycin was administered at a dosage of 5.7–10 mg/kg/day. Vancomycin was administered at a dosage of 15–20 mg/kg every 12 h to every 8 h. All studies were included if they reported one or more of the following outcomes: bacteremia recurrence, in-hospital mortality, duration of bacteremia, and adverse events. Studies with a population of participants who were younger than 18 years were excluded.

### 3.4. Definitions and Outcomes

The primary outcome was in-hospital mortality. In-hospital mortality was the death rate from all causes of death before patient discharge. The secondary outcomes were bacteremia recurrence, duration of bacteremia, and adverse events. Bacteremia recurrence was defined as at least one positive blood culture for MRSA seven or more days after the initial microbiological cure. The duration of bacteremia cure was defined as the number of days between the first positive blood culture and the first negative blood culture without a subsequent positive blood culture within 72 h of the negative blood culture. The adverse event data recorded were the risk of discontinuing due to adverse events, the incidence of serious adverse events, and some common events, such as diarrhea, nausea, headache, constipation, and seizure.

### 3.5. Quality Assessment and Statistical Analysis

The methods of quality assessment of the included studies and the statistical analysis of the data were the same as those used in a previous study [35].

## 4. Discussion

The current meta-analysis of six studies provides evidence that the in-hospital mortality rates, duration of bacteremia, and incidence of adverse drug events were not significantly different between the combination therapy and standard care of MRSA bacteremia. Combination therapy has a lower rate of bacteremia recurrence. In addition, daptomycin or vancomycin plus ceftaroline combination therapy, compared with monotherapy, did not reduce the risk of in-hospital mortality in the treatment of MRSA bacteremia patients.

The standard-of-care therapy for MRSA bacteremia is associated with high morbidity and mortality. Medical experts should explore the use of two antibiotics in combination. Four meta-analyses of combination therapy in the treatment of MRSA bacteremia were published in the literature [36,37,38,39]. Ye et al. (2020) included six studies of vancomycin combined with beta-lactam antibiotics and showed that there was significantly reduced persistent bacteremia and a shortened duration of bacteremia in the combination therapy group. There was no statistically significant difference in the incidence of nephrotoxicity, 30-day mortality, MRSA-related mortality, or bacteremia relapse between the two groups [36]. The study of Wang et al. (2020) included 15 studies of patients treated with daptomycin or vancomycin in combination with beta-lactam antibiotics and showed that the combination therapy significantly reduced the bacteremia recurrence and persistent bacteremia and shortened the duration of bacteremia. There was no statistically significant difference in the risk of crude mortality between the two groups. However, a subgroup analysis of three studies showed that the combination of daptomycin plus beta-lactam antibiotics could reduce the risk of crude mortality [37]. The study by Kale-Pradhan et al. (2020) included nine studies of patients treated with daptomycin or vancomycin in combination with beta-lactam antibiotics and demonstrated that the combination therapy was associated with significantly lower rates of bacteremia relapse and persistent bacteremia. Mortality was not significantly different between the two groups [38]. Yi et al. (2021) included 13 studies of patients treated with daptomycin or vancomycin in combination with beta-lactam antibiotics and found no statistically significant difference in 30-day mortality, in-hospital mortality, or mortality within 60–90 days between the two groups. Combination therapy is associated with a shorter duration of bacteremia, a lower risk of persistent bacteremia, and a lower risk of bacteremia recurrence within 60–90 days [39]. The previous four meta-analyses showed that adding a beta-lactam antibiotic to vancomycin or daptomycin decreased the recurrence of bacteremia and shortened the bacteremia duration in the treatment of patients with MRSA bacteremia. There was no evidence that combination therapy could reduce the risk of MRSA bacteremia mortality. We only included ceftaroline in combination with vancomycin or daptomycin versus vancomycin or daptomycin in the current meta-analysis. Our results were the same as those of the previous four meta-analyses. Combination therapy in the treatment of MRSA bacteremia did not reduce the risk of mortality, implying no significant benefit for patients with MRSA bacteremia. 

Lodise et al. suggested that the administration of MRSA bacteremia therapy within the first 24–48 h was strongly related to clinical outcomes [40]. Studies have shown that high-risk MRSA bacteremia patients benefit the most from combination therapy when it is administered early in the treatment course (within 72 h) [41]. It is important to initiate combination therapy early in the treatment of MRSA bacteremia, and it should be initiated within the first 72 h of onset, ideally within the first 24 h to prevent complications from persistent bacteremia [42,43,44]. Many studies have stressed that administering initial therapy within 72 h of the index culture is strongly related to MRSA bacteremia mortality. We analyzed two studies in the current meta-analysis, namely, the study of McCreary et al. and the study of Geriak et al. [30,31]. In the study of McCreary et al., the patients receiving daptomycin with ceftaroline combination therapy for MRSA bacteremia had a lower all-cause mortality at day 30 than those who received standard-of-care monotherapy. A subgroup analysis showed that there was a numerically lower mortality in the patients who received daptomycin plus ceftaroline within 72 h of the index culture. The study suggested that daptomycin and ceftaroline may have mortality benefits when initiated early for MRSA bacteremia [31]. In the study of Geriak et al., vancomycin or daptomycin was used as a monotherapy, and a regimen of daptomycin plus ceftaroline was used as a comparator for the initial treatment of MRSA bacteremia. That study observed an unanticipated in-hospital mortality difference of 0% (0/17) for combination therapy and 26% (6/23) for monotherapy, causing the early termination of the study [30]. That study also showed that daptomycin and ceftaroline combination therapy have mortality benefits as initial therapy within 72 h of the index culture in the treatment of MRSA bacteremia. The current meta-analysis recommended initial combination therapy with ceftaroline for MRSA bacteremia rather than ceftaroline as salvage therapy because initial combination therapy with ceftaroline may reduce the risk of MRSA bacteremia mortality. The current challenges with vancomycin or daptomycin plus ceftaroline combination in the treatment of MRSA bacteremia include a lack of ceftaroline data for the treatment of MRSA bacteremia, and a lack of data on the efficacy and safety of initial combination therapy for MRSA bacteremia. We call into question whether combination therapy works. Which combinations are best for MRSA bacteremia patients? What is the appropriate duration of combination therapy? Is combination therapy necessary for the entire course of treatment? Whether a de-escalation treatment regimen is considered a reasonable alternative to long-term combination therapy in patients with an early clinical response remains to be determined. In the future, four issues need to be explored by medical experts, which are as follows: 1. Initial and early combination therapy with daptomycin or vancomycin plus ceftaroline may be beneficial for mortality in MRSA bacteremia patients. Blinded, randomized, prospective studies are needed to confirm the efficacy and safety of combination therapy in MRSA bacteremia patients. 2. Appropriate dosing strategies for daptomycin or vancomycin plus ceftaroline combination therapy have not been determined. 3. Combination therapy is not necessary for the entire course of treatment. If de-escalation therapy is considered a reasonable alternative to long-term combination therapy in patients with an early clinical response, further investigation is warranted to determine the optimal timing of de-escalation. These drugs, including their dosage regimen and duration of therapy, are optimal for de-escalation therapy. 4. In the study of Geriak et al., a higher mortality was seen in patients with serum interleukin-10 concentrations >5 pg/mL [30]. The authors recommend the use of biomarkers as potential risk indicators for the administration of combination therapy in high-risk patients. Biomarkers related to MRSA bacteremia are a new, attractive area that is worth exploring. The medical community urgently needs advanced knowledge of biomarkers related to MRSA bacteremia to guide clinical decision-making and the management of MRSA bacteremia patients. 

Limitations: Few RCTs have explored this issue. We included the findings of observational studies in the current meta-analysis. All of the included studies had a high risk of bias in the current meta-analysis. In addition, the number of included studies and the number of populations were very small, which was a limitation of this meta-analysis. However, vancomycin or daptomycin plus ceftaroline combination in the treatment of MRSA bacteremia constitutes a promising possibility for reducing its mortality. We expect that there will be further studies that will explore this issue so as to provide a new way to treat MRSA bacteremia patients.

## 5. Conclusions

In the current meta-analysis, there was a trend that showed initial combination therapy with ceftaroline for MRSA bacteremia reducing the risk of MRSA bacteremia mortality. RCTs are needed to further study the role of initial combination therapy with daptomycin or vancomycin plus ceftaroline in the treatment of MRSA bacteremia.

## Figures and Tables

**Figure 1 antibiotics-11-01104-f001:**
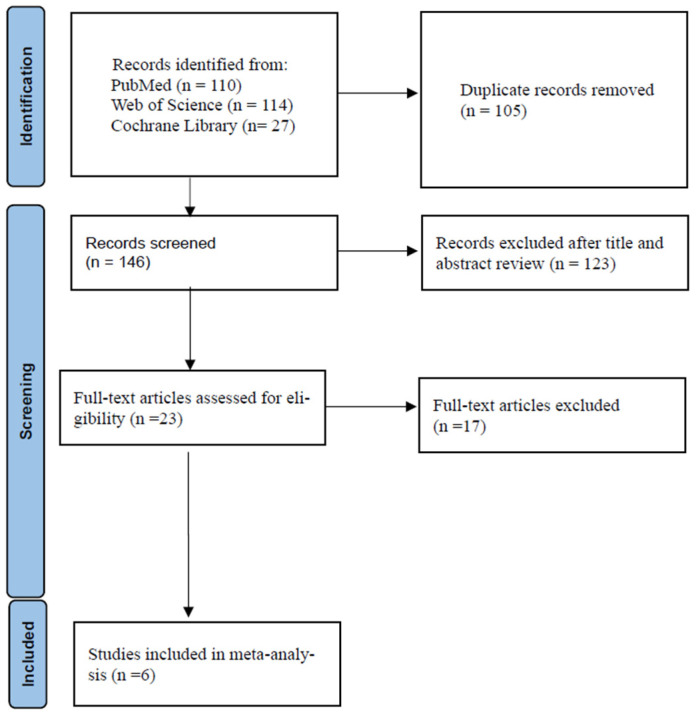
Flow diagram of the study selection process.

**Figure 2 antibiotics-11-01104-f002:**
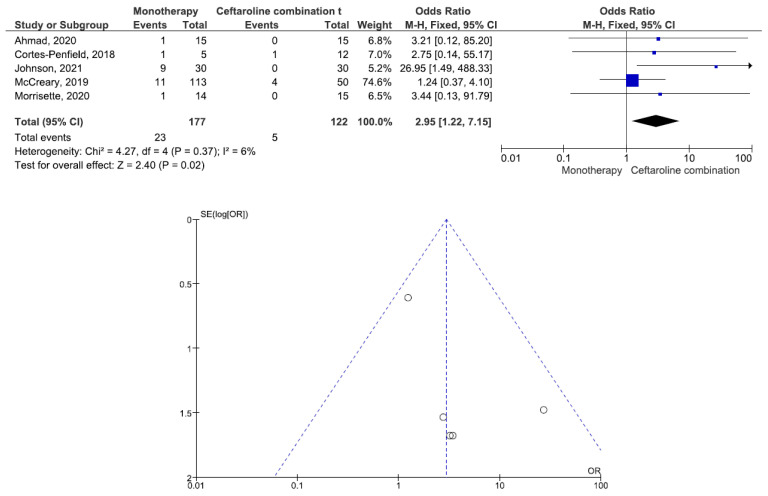
Forest plots and funnel plots for bcteremia recurrence between monotherapy and combination therapy in the treatment of methicillin-resistant *Staphylococcus aureus* bacteremia.

**Figure 3 antibiotics-11-01104-f003:**
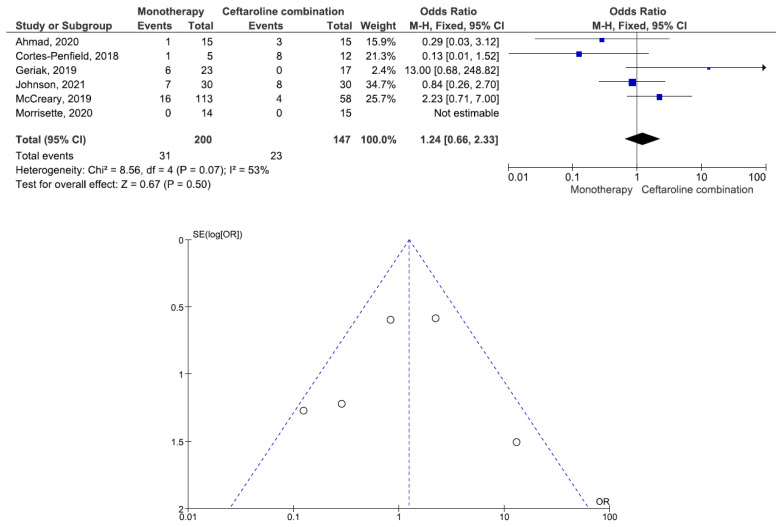
Forest plots and funnel plots for in-hospital mortality between monotherapy and combination therapy in the treatment of methicillin-resistant *Staphylococcus aureus* bacteremia.

**Figure 4 antibiotics-11-01104-f004:**
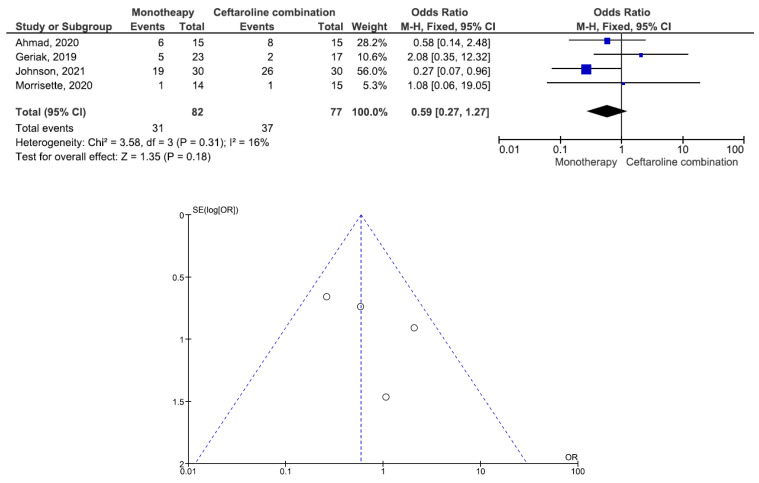
Forest plots and funnel plots for adverse events between monotherapy and combination therapy in the treatment of methicillin-resistant *Staphylococcus aureus* bacteremia.

**Table 1 antibiotics-11-01104-t001:** Characteristics of the included studies.

Author/Year	Region	Study Type	Initial or Salvage Therapy	Primary Infection Site	Drug Dosage	Duration of Bacteremia
Cortes-Penfield N, 2018 [29]	USA	RET	Salvage	Multiple	DAP: 5.7–13.8 mg/kg/day CPT: no data	MON: 12.0 days COM: 15.2 days
Geriak M, 2019 [30]	USA	RCT	Initial	Multiple	DAP: 6–8 mg/kg/day CPT: 600 mg q8h	MON: 3 days COM: 3 days
McCreary EK, 2019 [31]	USA	RET	Initial and salvage	Multiple	DAP: 8.2 mg/kg/day CPT: 600 mg q8h	MON: 4.8 days COM: 3.3 days
Morrisette T, 2020 [32]	USA	RET	Salvage	Multiple (Most IE)	DAP: 8.4–9.9 mg/kg/day CPT: 600 mg q8h	MON: 6.7 days COM: 7.6 days
Ahmad O, 2020 [33]	USA	RET	Salvage	IE, brain abscess, OMS	DAP: 8–10 mg/kg/day VAN:15–20 mg/kg, q12h-q8h CPT: 600 mg q12h-q8h	MON: 6.0 days COM: 6.0 days
Johnson TM, 2021 [34]	USA	RET	Salvage	Multiple	DAP: 100 mg/kg/day CPT: 600 mg q8h	MON: 5.0 days COM: 9.0 days

RCT: randomized controlled trial; RET: retrospective study; MON: monotherapy; COM: combination therapy; VAN: vancomycin; DAP: daptomycin; CPT: ceftaroline; IE: infective endocarditis; OMS: osteomyelitis; USA: United States of America.

## Data Availability

The datasets generated during and/or analyzed during the current study are not publicly available but are available from the corresponding author on reasonable request.

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
