# Peer review of "Comparing the Outcomes of Ceftaroline plus Vancomycin or Daptomycin Combination Therapy versus Vancomycin or Daptomycin Monotherapy in Adults with Methicillin-Resistant Staphylococcus aureus Bacteremia—A Meta-Analysis"

_antibiotics, 2022, doi:10.3390/antibiotics11081104_

Round 1
Reviewer 1 Report
1.- The format and order of the template must be respected. Example: The results should be written before the methodology, and the sections must be numbered.
2.- The titles of the figures must be written at the bottom of these (Figure 2)
3.- Line 19, 24: Staphylococcus aureus its not in italic
4.- Line 36-37: The authors says: "Treatment with vancomycin will clinically or microbiologically fail in the majority of invasive MRSA infections". More clinical and bibliographic evidence is needed to support this statement. In daily clinical practice this is not the case.
5.- Line 52-54: The authors says: "It is also active against common gram-negative pathogens such as Escherichia coli, Klebsiella pneumoniae, and Haemophilus influenzae; it is not active against extended spectrum β-lactamase-producing gram-negative organisms". It should be specified that the activity against Gram negative pathogens is in vitro.
6.- The final conclusions of the work must be in line with the conclusion of the abstract.
7.- Bacterial species names should be in italics. Review the text with emphasis on the bibliography.
Author Response
1.The format and order of the template must be respected. Example: The results should be written before the methodology, and the sections must be numbered.
Reply: I correct it.
2.The titles of the figures must be written at the bottom of these (Figure 2)
Reply: I correct it.
3 Line 19, 24: Staphylococcus aureus its not in italic
Reply: I correct it.
4.Line 36-37: The authors says: "Treatment with vancomycin will clinically or microbiologically fail in the majority of invasive MRSA infections". More clinical and bibliographic evidence is needed to support this statement. In daily clinical practice this is not the case.
Reply:
- Treatment with vancomycin will clinically or microbiologically fail in invasive MRSA infections.
- Cites five references
1.Forstner C, Dungl C, Tobudic S, Mitteregger D, Lagler H, Burgmann H. Predictors of clinical and microbiological treatment failure in patients with methicillin-resistant Staphylococcus aureus (MRSA) bacteraemia: a retrospective cohort study in a region with low MRSA prevalence. Clin Microbiol Infect. 2013 Jul;19(7):E291-7. doi: 10.1111/1469-0691.12169.
2.Lin SH, Liao WH, Lai CC, Liao CH, Tan CK, Wang CY, Huang YT, Hsueh PR. Risk factors for mortality in patients with persistent methicillin-resistant Staphylococcus aureus bacteraemia in a tertiary care hospital in Taiwan. J Antimicrob Chemother. 2010 Aug;65(8):1792-8. doi: 10.1093/jac/dkq188.
3.Lodise TP, Graves J, Evans A, Graffunder E, Helmecke M, Lomaestro BM, Stellrecht K. Relationship between vancomycin MIC and failure among patients with methicillin-resistant Staphylococcus aureus bacteremia treated with vancomycin. Antimicrob Agents Chemother. 2008 Sep;52(9):3315-20. doi: 10.1128/AAC.00113-08
4.Yoon YK, Kim JY, Park DW, Sohn JW, Kim MJ. Predictors of persistent methicillin-resistant Staphylococcus aureus bacteraemia in patients treated with vancomycin. J Antimicrob Chemother. 2010 May;65(5):1015-8. doi: 10.1093/jac/dkq050
5.Gould IM. Treatment of bacteraemia: meticillin-resistant Staphylococcus aureus (MRSA) to vancomycin-resistant S. aureus (VRSA). Int J Antimicrob Agents. 2013 Jun;42 Suppl:S17-21. doi: 10.1016/j.ijantimicag.2013.04.006.
- Line 52-54: The authors says: "It is also active against common gram-negative pathogens such as Escherichia coli, Klebsiella pneumoniae, and Haemophilus influenzae; it is not active against extended spectrum β-lactamase-producing gram-negative organisms". It should be specified that the activity against Gram negative pathogens is in vitro.
Reply: In vitro activity of ceftaroline is also active against common gram-negative pathogens such as Escherichia coli, Klebsiella pneumoniae, and Haemophilus influenzae; it is not active against extended spectrum β-lactamase-producing gram-negative organisms.
- The final conclusions of the work must be in line with the conclusion of the abstract.
Reply: There was a trend for initial combination therapy with ceftaroline for MRSA bacteremia in reducing the risk of MRSA bacteremia mortality in the current meta-analysis. RCTs controlled trials are needed to further study the role of initial combination therapy with daptomycin or vancomycin plus ceftaroline in the treatment of methicillin-resistant Staphylococcus aureus bacteremia.
- Bacterial species names should be in italics. Review the text with emphasis on the bibliography.
Reply: I correct it.
Reviewer 2 Report
The purpose of this research was the meta-analysis in order to compare the clinical outcome of methicillin-resistant Staphylococcus aureus bacteremia between patients treated with daptomycin or vancomycin plus ceftaroline combination therapy and daptomycin or vancomycin monotherapy. These are undoubtedly very important studies. However, four meta-analyses of combination therapy in the treatment of MRSA bacteremia were published in the literature. The results are very similar. The most important conclusion of this study is the statement that: “Initial combination therapy with ceftaroline for the treatment methicillin-resistant Staphylococcus aureus bacteremia showed a trend of reduced risk of in-hospital mortality in the current meta-analysis”. The authors are aware of the fact that the further study are needed and make a conclusion: “Randomized controlled trials are needed to further study the role of initial combination therapy with daptomycin or vancomycin plus ceftaroline in the treatment of methicillin-resistant Staphylococcus aureus bacteremia”.
The “Efficacy and safety outcomes” is the weakest point of the work. Some information was presented in imprecise way. For Example: “There were 200 patients in the monotherapy group and 147 patients in the combination therapy group. Five studies involving 299 patients (177 receiving monotherapy, 122 receiving ceftaroline combination therapy) reported bacteremia recurrence”. I think that this section should be improved and rewritten.
Author Response
- Efficacy and safety outcomes
There were 200 patients in the monotherapy group and 147 patients in the combination therapy group. Five studies involving 299 patients (177 receiving monotherapy, 122 receiving ceftaroline combination therapy) reported bacteremia recurrence. There was a statistically significant difference in the bacteremia recurrence between the patients treated with monotherapy and those treated with ceftaroline combination therapy (OR=2.95, 95% CI= 1.22-7.15, P=0.02, I2=6%) (Figure 2). All five studies reported that the combination therapy group had less bacteremia recurrence than the monotherapy group. Six studies involving 347 patients (200 receiving monotherapy, 147 receiving ceftaroline combination therapy) reported in-hospital mortality. There was no statistically significant difference in the in-hospital mortality between the patients treated with monotherapy and those treated with ceftaroline combination therapy (OR=1.24, 95% CI= 0.66-2.33, P=0.50, I2=53%) (Figure 3). Three studies favored monotherapy and two studies favored combination therapy. Merrisette et al. found no deaths in either group [32]. We analyzed the two studies in which the combination therapy group had better in-hospital hospital mortality than the monotherapy group [30,31]. Geriak M et al. showed that combination therapy had mortality benefits as an initial therapy within 72 hours of the index culture in the treatment of MRSA bacteremia [30]. McCreary EK et al. found a lower all-cause mortality at day 30 in the combination therapy group than in those who received standard-of-care monotherapy, and there was lower mortality in the patients who received daptomycin plus ceftaroline within 72 hours of the index culture [31]. Four studies involving 159 patients (82 receiving monotherapy, 77 receiving ceftaroline combination therapy) reported adverse events. There was no statistically significant difference in the adverse events between the patients treated with monotherapy and those treated with ceftaroline combination therapy (OR= 0.59, 95% CI= 0.27-.1.27, P=0.18, I2=16%) (Figure 4). There were two cases of eosinophilic pneumonitis and three cases of elevated creatine phosphokinase in the combination therapy group. There was no case of eosinophilic pneumonitis but two cases of elevated creatine phosphokinase in the monotherapy group. Regarding the duration of bacteremia, four studies reported no statistically significant difference in the duration of bacteremia between the patients treated with monotherapy and those treated with ceftaroline combination therapy [29,30,32,33]. In the study of McCreary EK et al., the mean duration of bacteremia was 4.8 days in the standard of care group and 9.3 days in the daptomycin plus ceftaroline group (P <0.001); following the switch to daptomycin plus ceftaroline, the mean duration of continued bacteremia was 3.3 days for the combination therapy. The longer duration of bacteremia was related to daptomycin plus ceftaroline as a salvage therapy, and there was 6.0 mean bacteremia days before switching to combination therapy [31]. A study by Johnson TM et al. showed that the mean duration of bacteremia was 5.0 days in the standard of care group and 9.0 days in the daptomycin plus ceftaroline group (P =0.01). The total duration of bacteremia was significantly longer in the daptomycin plus ceftaroline group because the patients failed first-line therapy (after a median of 6 days) before daptomycin plus ceftaroline combination therapy. The longer duration of bacteremia was related to daptomycin plus ceftaroline as a salvage therapy [34]. We found that daptomycin plus ceftaroline as a salvage therapy was associated with a longer duration of bacteremia in both studies.
Reviewer 3 Report
The manuscript compared the clinical outcomes of in-hospital mortality, recurrence of MRSA bacteremia, duration of bacteremia, and adverse events between patients treated with daptomycin or vancomycin plus ceftaroline combination therapy and daptomycin or vancomycin monotherapy. The manuscript has potential; however, the results are inconsistent due to the low number of articles included in the study. Therefore, it is not suitable for publication in the present format.
Minor points
Lines 14: Put the abbreviation MRSA after the full name
Lines 19, 24, 27, 41/42, 135: Put the abbreviation MRSA instead of the full name
Line 29: Remove the italics from Methicillin-resistant
Line 41/42: Remove the capital letter from the beginning of words
Line 45: Remove the space before such
Lines 45, 54, 55, 255, 259: Standardize beta or β in all text
Line 79: Remove the capital letter “and”
Lines 109-110: On the topic of inclusion and exclusion criteria, why did the authors exclude studies with a population of participants under 18 years of age?
Line 123: This sentence is unnecessary “We recently published two meta-analyses in the literature”
Line 180: Table 1 is confusing and must be reviewed for presentation, format, captions, and footnotes.
Lines 220-228: The quality of figures 2, 3, and 4 must be improved. Where is the study of Geriak M in figure 2? And Cortes-Penfield N and McCreary EK in figure 4?
Lines 343-347: Revise all
Lines 356-499: All references must be revised
Author Response
Minor points
Lines 14: Put the abbreviation MRSA after the full name
Reply: I keep no abbreviation in the abstract part.
Lines 19, 24, 27, 41/42, 135: Put the abbreviation MRSA instead of the full name
Reply: I correct it
Line 29: Remove the italics from Methicillin-resistant
Reply: I correct it
Line 41/42: Remove the capital letter from the beginning of words
Reply: I correct it
Line 45: Remove the space before such
Reply: I correct it
Lines 45, 54, 55, 255, 259: Standardize beta or β in all text
Reply: I correct it
Line 79: Remove the capital letter “and”
Reply: I correct it
Lines 109-110: On the topic of inclusion and exclusion criteria, why did the authors exclude studies with a population of participants under 18 years of age?
Relay:
1.No study focus on population of participants under 18 years of age in the literature.
2.I am not a pediatric and I do not have the knowledge and experience to explored the pediatric MRSA bacteremia.
3.This meta-analysis is explored in adults (Comparing the outcomes of ceftaroline plus vancomycin or daptomycin combination therapy versus vancomycin or daptomycin monotherapy in adults with methicillin-resistant Staphylococcus aureus bacteremia—A meta-analysis)
Line 123: This sentence is unnecessary “We recently published two meta-analyses in the literature”
Reply: I delete it
Line 180: Table 1 is confusing and must be reviewed for presentation, format, captions, and footnotes.
Reply: I correct it
Lines 220-228: The quality of figures 2, 3, and 4 must be improved. Where is the study of Geriak M in figure 2? And Cortes-Penfield N and McCreary EK in figure 4?
Reply: 1.I correct it. 2. No report bacteremia recurrence in the study of Geriak M. 3. No report adverse events in the studies of Cortes-Penfield N and McCreary EK.
Lines 343-347: Revise all
Reply: All authors made substantial contributions to conception and design, acquisition of data, or analysis and interpretation of data; took part in drafting the article or revising it critically for important intellectual content; agreed to submit to the current journal; gave final approval of the version to be published; and agree to be accountable for all aspects of the work.
Lines 356-499: All references must be revised
Reply: I correct all references guided by instructions of authors.
Round 2
Reviewer 3 Report
The changes made by the authors in the manuscript meet the questions I have raised. And now I recommend its publication.
I call attention to a small detail that still needs to be corrected, but that does not interfere with the acceptance of the publication and can be done during the text editing. The abbreviation of the name of the bacterial species must occur after the first citation in full.
Author Response
1.I check it and correct it.
2.This article will be checked again before publication.